# AuNP Aptasensor for Hodgkin Lymphoma Monitoring

**DOI:** 10.3390/bios12010023

**Published:** 2022-01-04

**Authors:** Maria Slyusarenko, Sergey Shalaev, Alina Valitova, Lidia Zabegina, Nadezhda Nikiforova, Inga Nazarova, Polina Rudakovskaya, Maxim Vorobiev, Alexey Lezov, Larisa Filatova, Natalia Yevlampieva, Dmitry Gorin, Pavel Krzhivitsky, Anastasia Malek

**Affiliations:** 1Subcellular Technology Laboratory, Department of Hematology and Chemotherapy and Department of Radionuclide Diagnostics, N.N. Petrov National Medical Research Center of Oncology, 197758 St. Petersburg, Russia; slusarenko_masha@mail.ru (M.S.); shalaev.haemonc@gmail.com (S.S.); valitva.alina@gmail.com (A.V.); lidusikza@yandex.ru (L.Z.); niki2naden_ka@mail.ru (N.N.); oblaka12@mail.ru (I.N.); larisa_filatova@list.ru (L.F.); krzh@mail.ru (P.K.); 2The Faculty of Physics and Center for Molecular and Cell Technologies, Saint-Petersburg State University, 199034 St. Petersburg, Russia; vorobiev.maxim@rambler.ru (M.V.); alezov@gmail.com (A.L.); n.yevlampieva@spbu.ru (N.Y.); 3Center for Photonics and Quantum Materials, Skolkovo Institute of Science and Technology, 121205 Moscow, Russia; polinaru@list.ru (P.R.); d.gorin@skoltech.ru (D.G.)

**Keywords:** enzyme-mimetic nanoparticles, gold nanoparticles, aptamer, liquid biopsy, small extracellular vesicles, Hodgkin lymphoma, diagnostic, monitoring

## Abstract

A liquid biopsy based on circulating small extracellular vesicles (SEVs) has not yet been used in routine clinical practice due to the lack of reliable analytic technologies. Recent studies have demonstrated the great diagnostic potential of nanozyme-based systems for the detection of SEV markers. Here, we hypothesize that CD30-positive Hodgkin and Reed–Sternberg (HRS) cells secrete CD30 + SEVs; therefore, the relative amount of circulating CD30 + SEVs might reflect classical forms of Hodgkin lymphoma (cHL) activity and can be measured by using a nanozyme-based technique. A AuNP aptasensor analytics system was created using aurum nanoparticles (AuNPs) with peroxidase activity. Sensing was mediated by competing properties of DNA aptamers to attach onto surface of AuNPs inhibiting their enzymatic activity and to bind specific markers on SEVs surface. An enzymatic activity of AuNPs was evaluated through the color reaction. The study included characterization of the components of the analytic system and its functionality using transmission and scanning electron microscopy, nanoparticle tracking analysis (NTA), dynamic light scattering (DLS), and spectrophotometry. AuNP aptasensor analytics were optimized to quantify plasma CD30 + SEVs. The developed method allowed us to differentiate healthy donors and cHL patients. The results of the CD30 + SEV quantification in the plasma of cHL patients were compared with the results of disease activity assessment by positron emission tomography/computed tomography (PET-CT) scanning, revealing a strong positive correlation. Moreover, two cycles of chemotherapy resulted in a statistically significant decrease in CD30 + SEVs in the plasma of cHL patients. The proposed AuNP aptasensor system presents a promising new approach for monitoring cHL patients and can be modified for the diagnostic testing of other diseases.

## 1. Introduction

The classical form of Hodgkin lymphoma (cHL) is a lymphoid malignancy with malignant cells (Hodgkin and Reed–Sternberg (HRS) cells) representing only a small percentage of a tumor’s volume; the tumor is mostly composed of non-neoplastic infiltrating cells. Due to this peculiarity of cHL, the genetic landscape of the disease has not yet been characterized in detail, and no circulating markers derived from HRS cells have been identified [1,2]. In recent years, the complex interplay between rare HRS tumor cells and their microenvironment implicating molecular mediators (cytokines, chemoattractant, and chemokines) and membrane vesicles has attracted significant interest [3]. Small extracellular vesicles (SEVs) are lipid-bilayer-bound vesicles released from living cells into the extracellular environment, which mediate cell-to-cell communication and are present in most of the biological fluids including plasma [4]. For instance, SEVs secreted by HRS cells into interstitial spaces have been detected in circulating plasma and could present a potential source of biomarkers [5].

Aptamers are single-stranded DNA (ssDNA) or RNA molecules capable of tightly binding to specific molecular targets [6]. A combinatorial technique, termed the systematic evolution of ligands by exponential enrichment (SELEX), has been developed to identify aptamers with an affinity to specific molecular targets [7]. For instance, ssDNA aptamers for cHL tumor cells were found using a cell-based SELEX protocol [8]. Aptamers capable of capturing and detecting SEVs have been considered to be a promising type of biosensors, or aptasensor [9]. Regardless of the target specificity, aptasensors can be classified based on sensing technologies as follows: fluorescence, electrochemistry, luminescence, colorimetry, surface-enhanced Raman scattering, surface plasmon resonance, and giant magnetoresistance. The limit of SEV’s detection for some of these aptasensors reaches 10^2^ to 10^3^ vesicles/mL (Table 2 in [9]), suggesting a high sensitivity and great diagnostic potential of aptamer-based techniques. Colorimetric aptasensors employ the ability of ssDNA aptamers to reversibly modulate either the colloidal stability or enzyme-mimetic activity of metal nanoparticles. For instance, a colorless suspension of gold nanoparticles (AuNPs) stabilized by aptamers was turned dark after adding SEVs due to AuNP aggregation [10]. The aptamer-enhanced peroxidase activity of various nanomaterials such as Fe_3_O_4_ nanoparticles [11], g-C_3_N_4_ nanosheets [12], and carbon nanotubes [13] has been decreased due to the addition of SEVs, and has been described as a method for the sensitive quantification of SEVs. 

However, the phenomenon of enzyme-mimetic activity of nanomaterials has still not been adequately studied and its manifestations are not always predictable. In contrast to the above-mentioned reports describing aptasensors for SEV detection, Wang et al. demonstrated that ssDNA was adsorbed on the surface of AuNPs via nonspecific electrostatic interactions and thereby blocked the peroxidase activity of nanoparticles [14]. In this system, AuNPs complexed with ssDNA kanamycin-specific aptamers did not exhibit peroxidase activity, whereas the target-induced replacement of aptamers exposed the surface of AuNPs and recovered the peroxidase activity. Apparently, the enzyme-mimetic activity of NPs is defined by many inherent factors including chemistry, geometry, and surface characteristics [15], and needs to be experimentally evaluated in each specific conditions. 

The goal of the present study was to explore the sensitivity of a AuNP aptasensor for the quantification of CD30 + SEVs in the plasma of cHL patients and to test the disease-indicative potential of such quantification. With this purpose, we evaluated the suppressing effects of ssDNA aptamer adsorption on the peroxidase activity of AuNPs and created a AuNP aptasensor analytic system based on the phenomenon of SEV-induced recovery of AuNP peroxidase activity. The sensitivity of the AuNP aptasensor appeared to be sufficient to discriminate donors and cHL patients; the results of HRS-cell-derived SEV quantification in the plasma of cHL patient correlated with the results of PET-CT scanning. Moreover, the AuNP aptasensor allowed us to detect a reduction in HRS-cell-derived SEVs in the plasma of cHL patients after two cycles of chemotherapy. Thus, the proposed AuNP aptasensor presents a promising technology that can be further optimized and evaluated in clinical settings as a new approach for cHL monitoring and the early detection of disease recurrence. 

## 2. Materials and Methods

### 2.1. Hypothesis and Plan of Study 

The hypothesis of our study was based on the following assumptions.

First, high expression of CD30 (TNFSF8 receptor) is a well-established diagnostic marker of Hodgkin and Reed–Sternberg (HRS) cells. Most HL cell lines (L540, L428, KM-H2, and L1236) express CD30, whereas SEVs secreted by these cells are CD30-positive [16,17]. If the development of cHL is associated with the appearance of HRS-cell-derived SEVs in circulation, the vesicles positive for CD30 and/or other cHL-specific molecules might serve as disease-indicative markers. 

Second, the peroxidase-mimetic activity of AuNPs can be reversibly suppressed by the attachment of ssDNA onto the surface of particles [14]. The replacement of ssDNA might expose the surface of AuNPs and recover their peroxidase activity. Thus, reversible enzymatic activity enables the biosensor to function. 

Third, DNA aptamers selected for the specific binding of surface markers of cHL cells will have dual properties, i.e., a nonspecific attachment to the surface of AuNPs as well as a highly affine interaction with cHL-specific markers of the surfaces of HRS-cell-derived SEV. For instance, the peroxidase activity of AuNPs can be reversibly switched off and on by electrostatic attachment and the CD30-induced detachment of CD30-specific aptamers, and can therefore reflect the amount of CD30 + SEVs in the reaction mixture. 

Thus, AuNPs and HRS-specific aptamers might form a biosensor (AuNP aptasensor) sufficiently sensitive for the accurate quantification of HRS-cell-derived SEVs in plasma. In order to prove this hypothesis, we (i) first, characterized components of the analytics system; (ii) then, optimized the analytic conditions using aptamers binding nonspecific SEV marker CD63; (iii) and finally, evaluated the performance of AuNP aptasensor using CD30-specific aptamers and plasma samples of healthy donors and cHL patients. 

### 2.2. Aurum Nanoparticle (AuNP) Synthesis

The synthesis of aurum nanoparticles (AuNPs) was carried out according to the method previously described in [18]. Briefly, a solution of 75 mg H [AuCl_4_]·3H_2_O in 110 mL of distilled water was placed in a 250 mL round flask equipped with a reverse solution and the solution was brought to a boil. Then, 26.25 mL of 1% sodium citrate solution was quickly added and boiling was continued for one hour. Afterwards, the AuNPs were cooled to room temperature with stirring. The AuNPs were characterized using a Zetasizer Nano ZS and Nanosight NS300 analyzer (both from Malvern Panalytical, Malvern, USA) according to the manufacturer’s instructions and visualized by scanning electron microscopy (SEM), as described below. 

### 2.3. Isolation of Small Extracellular Nanovesicle from Plasma (SEVs)

Blood samples from cHL patients and healthy donors were obtained by blood draws. The blood was collected in EDTA-coated tubes, the plasma was immediately separated by centrifugation (15 min at 3000× *g* at 4 °C), aliquoted, and stored at −80 °C. Before use, the plasma was slowly thawed at 4 °C. In order to remove cellular detritus, the plasma was centrifuged, sequentially, for 10 min at 400× *g*, 10 min at 800× *g*, 10 min at 1500× *g*, and 30 min at 17,000× *g*. Each time, the supernatant was carefully replaced in a new tube. After the last centrifugation, the supernatant was filtered through a 0.2 µm PES syringe filter to obtain pellet pure plasma (PPP). Plasma treated in this way was used in all further experiments.

The small extracellular nanovesicles (SEVs) were isolated from the PPP using a two-polymer system, previously described in [19]. The polymers, polyethylene glycol (PEG, 20 kDa), and dextran (DEX, 450–650 kDa) (both from Sigma-Aldrich, St. Louis, MO, USA), were dissolved in PBS at concentrations of 30% and 35%, respectively. Then, 50 μL of PEG solution and 150 μL of DEX solution were added to 1.3 mL of plasma to obtain the final concentrations, i.e., 1% of PEG and 3.5% of dextran. The same quantities of the polymer solutions were added to 1.3 mL of phosphate-buffered saline (PBS) to form protein-depleted solutions (PDSs). The solutions were well mixed for 30 min at 4 °C by vortexing, and then centrifuged for 10 min at 1000× *g* to speed up partition of the polymer solutions into the lower phase (LP) and upper phase (UP). To deplete the LP with plasma protein, the UP was removed and replaced with PDS. The solution was mixed and reseparated again by centrifugation at 1000× *g* for 10 min. The UP was removed, whereas the LP containing SEVs was dissolved in PBS, up to 100 μL, and used for the following experiments. 

### 2.4. Scanning Electron Microscopy (SEM) and Transmission Cryo-Electron Microscopy (Cryo-TEM)

The morphology of aurum nanoparticles (AuNPs) was studied by scanning electron microscopy (SEM) with a Zeiss Supra 40 VP microscope (Carl Zeiss Microscopy GmbH, Oberkochen, Germany). 

The SEVs isolated from the plasma were visualized by transmission cryo-electron microscopy (cryo-TEM). Carbon-film-coated Lacey carbon support copper grids (TEM-LC200CU25, Sigma-Aldrich, St. Louis, MO, USA) were treated (15 s, O_2_/H_2_) on a hydrophilic cleaning device Jeol EM-39010. A suspension of SEVs (4 μL) was added to a grid at a relative humidity of 90% and temperature of 20 °C. The excess suspension was blotted on filter paper, and the obtained film was vitrified in liquid ethane with an automatic fast sample freezer (Leica EM GP). The samples were imaged at −170 °C on a Gatan 914 cryo-holder for cryo-TEM imaging on a JEM-2100 transmission electron microscope (Jeol, Tokio, Japan). Images were taken using a Gatan Ultrascan 4000 camera at a magnification of either 30,000× or 15,000×.

### 2.5. Nanoparticle Tracking Analysis (NTA)

NTA was used to determine the size and concentration of the SEVs isolated from plasma. The samples were dissolved in PBS, 1:100. The measurements were carried out on a Nanosight NS300 analyzer (Malvern, PA, USA). The spectra were processed using Nanosight NTA 3.2 Software. The settings were: camera level 14, shutter slider 1259, slider gain 366, and threshold level 5. Each sample was pumped through the analyzer observation chamber and 5 measurements were taken at different microvolumes of the same sample. Each measurement lasted 60 s, which corresponded to 1498 frames. The results of 5 measurements were averaged. The experimental data were analyzed with NanoSight NTA 3.2 software.

### 2.6. Bead-Assisted Flow Cytometry

The exosomal markers (CD63 and CD9) on the surfaces of SEVs were assayed using an Exo-FACS kit (HansaBioMed Life Science, Tallinn, Estonia), according to the protocol. SEVs immobilized on the surface of the latex beads were labeled with monoclonal antibodies to tetraspanins CD63 or CD9, conjugated with FITC or PerCP-Cy5.5, respectively. The flow cytometry data were obtained on a CytoFLEX analyzer (Beckman Coulter, Brea, CA, USA) equipped for multi-parametric and multicolor analysis, including an argon laser set at 488 nm for measurements of forward light scatter (FSC) and orthogonal scatter (SSC). The complexes assembled without SEVs were used as negative controls. The data were analyzed with CytExpert Acquisition and Analysis Software (Beckman Coulter, Brea, CA, USA). 

### 2.7. Interaction of AuNPs, ssDNA Aptamers, and Extracellular Nanovesicles (SEVs)

To modulate the peroxidase activity of AuNPs, we used single-stranded ssDNA oligomers (aptamers): CAC CCC ACC TCG CTC CCG TGA CAC TAA TGC TA (CD63) [20]: length 32 bp, GC 59.4%, ACT GGG CGA AAC AAG TCT ATT GAC TAT GAG (CD30) [21]: length 30 bp, GC 43.3%, TAA CTT TGA AGG AAA GGC TAC AAA CTC TTC (PS1NP) [8]: length 30 bp, GC 36.7%, AGT GGT CTA ACT ACA CAT CCT TGA ACT ACG (SCR): length 30 bp, GC 43.3%, all purchased from Evrogen Ltd. (Moscow, Russia). As expected, the secondary structures of these molecules mediated their highly affine binding with specific markers CD63 [20], CD30 [21], and unidentified marker (PS1NP) of HL cells [8]. Scramble (SCR) oligomers with lengths and GC contents equivalent to CD30 aptamer were used as negative control. To form AuNP–ssDNA complexes, a constant amount of AuNP suspension (10 μL of suspension at a concentration of 2 × 10^15^ particles/mL) was mixed with 2.5 μL of aptamer solution at the desired concentration, and incubated for 30 min at 4 °C. Subsequently, the hydrodynamic radius of particles was assayed by dynamic light scattering (DLS), whereas the peroxidase activity of AuNPs was evaluated after adding 3,3′,5,5′-tetramethylbenzidine (10 μL) to the resulting solution and incubating for 18 min at 37 °C in the dark. We utilized a ready-to-use 3,3′,5,5′-tetramethylbenzidine (TMB) substrate solution (Xema Co. Ltd., Moscow, Russia). After incubation, the suspensions were centrifuged (6000× *g*, 3 min), the supernatants were carefully transferred to the wells of a 384-well plate, and the absorption spectra were measured immediately. 

The interactions between AuNP–ssDNA complexes and SEVs were studied by mixing the AuNP–DNA complexes (12.5 μL) and SEV (1 μL) suspensions at the desired concentrations. The mixtures were incubated for 30 min at 4 °C to release AuNPs and to recover their peroxidase activity. The hydrodynamic radii of particles in the resulting suspension and peroxidase activity of AuNPs were assayed, as described above. 

### 2.8. Spectrophotometry

Spectra registration was carried out via a Varioskan LUX Multimode Microplate Reader (Thermo Fischer Scientific, Waltham, MA, USA). Distilled water was used as a blank. The absorbance spectra were investigated in diapason from 200 to 800 nm. The results were processed in OriginPro software (OriginLab Corporation, Northampton, MA, USA). 

### 2.9. Dynamic Light Scattering 

The DLS experiments were carried out using “PhotoCor Complex” apparatus (Photocor Instruments Inc., Moscow, Russia). The device included a digital correlator (288 channels, 10 ns), a standard goniometer (10°–150°), and a thermostat with temperature stabilization of 0.05 °C. A single-mode linear polarized laser (λ_0_ = 654 nm) was used as an excitation source; the experiments were carried out at scattering angles (θ) ranging from 30° to 130°. Autocorrelation functions of scattered light intensity were processed using the inverse Laplace transform regularization procedure incorporated in DynaLS software (provided by Photocor Instruments Inc., Moscow, Russia), which provides distributions of scattered light intensities by relaxation times ρ(τ). The dependence of 1/τ (where τ is the position of a maximum of the ρ(τ) distribution) on the scattering vector squared was calculated as follows:q^2^ = (4πn/λ_0_sin(θ/2))^2^, (1)
where n is the refractive index of the solvent. For all studied samples, this was a straight line passing through the origin, indicating the diffusional character of the observed processes (1/τ = Dq^2^) [22]. The hydrodynamic radius, Rh, was calculated using the Stokes–Einstein equation [23] as follows:R_h_ = (k_B_ T)/(6πη_0_D), (2)
where k_B_ is the Boltzmann constant, T is the absolute temperature, and η_0_ is the solvent viscosity.

### 2.10. Patients

The study involved patients with a histologically confirmed classical form of HL (*n* = 10) and healthy donors (*n* = 10). All participants signed an informed consent form. The median age of patients was 34 years, including six males and four females. A lymph node biopsy with morphological and immune-histochemical (IHC) examination was performed for all patients, revealing three cases of mixed cellularity cHL and seven cases of nodular sclerosis cHL. Then, 2-deoxy-[18F]-fluoro-D-glucose positron emission tomography and computed tomography (18FDG-PET/CT) were performed to evaluate disease activity. The patients were staged on the bases of the PET-CT data and clinical status. The individual characteristics of patients, including systemic symptoms (B symptoms) and treatment tolerance expressed by ECOG scores, are presented in Table 1. The control group included healthy donors of corresponding ages and genders.

The cHL patients were all treated with standard regimes of chemotherapy with doxorubicin, bleomycin, vinblastine, and dacarbazine (ABVD). The effect of two cycles of chemotherapy was evaluated by PET-CT scanning. Briefly, scanning was performed after the intravenous administration of 250–540 Mbq of 18F-FDG in accordance with the standard protocol and parameters (CT 80 mA, 120 kV, no contrast enhancement, 3–4 min/bed-step of 15 cm). The images were processed using the software Syngo.via (Siemens Healthineers, Erlangen, Germany). In order to determine the main parameters, i.e., MTV and TLG, we used the special tool multi-foci segmentation (MFS) that automatically registered/delineated contours around each metabolically active focus as the sum of all voxels within VOI after determining the SUV background reference over the liver [24]. The threshold of FDG uptake was 41% of the SUV maximum inside, as recommended by the European Association of Nuclear Medicine. In all cases, contours were manually corrected to evade inclusion areas of physiological uptake. In addition, focuses that had an FDG uptake less than in the liver but estimated as malignant (e.g., small lesion size), were also drawn. At a median of 4–6 weeks after completion of the second cycle of chemotherapy, all patients underwent the second PET-CT scan under the same parameters as at baseline staging. A visual analysis was performed, using a five-point scale (Deauville criteria) recommended for treatment evaluation [25,26]. The PET-CT scans were interpreted by two experienced nuclear medicine physicians (A.V. and P.K.). Table 1 details the results of the PET-CT data evaluation (MTV and TLG) performed before and after therapy, as well as the results of the treatment response assessment with the Deauville scale. 

### 2.11. The Protocol of CD30 + SEV Quantification in Plasma Samples

In order to quantify CD30 + SEVs derived from CD30 + cells, the total population of plasma SEV was isolated from pellet pure plasma (1.3 mL) by the two-polymer system, as described in Section 2.3. The AuNP aptasensor was modified using the ssDNA aptamer with previously reported affinity to surface markers of HL cells. Thus, a highly affine interaction of the CD30 aptamer with the CD30 receptor was reported [21]. The PS1NP-aptamer was described to specifically bind cultured HL cells (HDLM2 and KMH2); however, it reacted with anaplastic large cell lymphoma with less affinity [8]. On the basis of the performed experiments, we optimized the procedure of the analysis. It included three steps of incubations, schematically presented in Figure 1. 

The AuNPs (2 × 10^15^ particles/mL, 10 μL) were mixed with aptamers (75 mM, 2.5 μL) to form the AuNP aptasensor (Figure 1A). Interactions of CD30-specific ssDNA aptamers with the surface of the AuNPs resulted in the complete silencing of their peroxidase activity. Then, the suspension of plasma SEVs (1 μL) was added to the reaction mixture and carefully mixed (Figure 1B). Incubation of the AuNP aptasensor with SEVs resulted in the replacement of CD30-specific ssDNA aptamers from the AuNP surface to CD30 on the SEV membrane, and peroxidase activity of the NPs was partially recovered. At the final stage, the TMB solution (10 μL) was added to the reaction mixture, the mixture was incubated, centrifuged (6000× *g*, 3 min) and the color reaction activity of TMB peroxidation was assayed by spectrophotometry of the supernatant (Figure 1C). Overall analysis took less than 2 h. 

## 3. Results

### 3.1. Components of the Analytic System

#### 3.1.1. Gold Nanoparticle (AuNP) Synthesis and Characteristics

The AuNPs were synthesized as described in Section 2.1. Dispersion stability of AuNPs was mediated by citrate solution, as shown schematically in Figure 2A. The AuNPs were characterized in terms of shape, size/hydrodynamic radius, ζ-potential, and concentration. 

The AuNPs had an almost-round form and diameters about 12 ± 2 nm, according to the TEM data (Figure 2B). The ζ-potential on the particle surface was −25 mV, as shown in Figure 1C, which indicated intermediate dispersion stability of the AuNPs. The hydrodynamic radius was measured by dynamic light scattering (DLS) using an indirect size calculation with the Stokes–Einstein formula. The hydrodynamic radii of AuNPs were 7 ± 2 nm (Figure 2D), which corresponded to 14 nm particle diameters. The slight discrepancy between the TEM and DLS data (12 nm vs. 14 nm) was acceptable due to methodological features. Additionally, the concentration of particles could be approximated from the gold concentration used for synthesis (260 mg ± 4 mg/mL) and was measured by nanoparticle tracking analysis (2 × 10^15^ particles/mL). The enzyme-mimetic activity of metal NPs has been reported previously [27]; therefore, the AuNPs with described characteristics were considered to be suitable for the next experiments. 

#### 3.1.2. Plasma SEV Isolation and Characterization 

In order to set up the SEV quantification system, the total population of plasma SEVs was isolated using a two-phase polymer system, described recently in [19]. This approach allowed us to isolate plasma SEVs quickly and efficiently. The SEVs were isolated from 1.3 mL of pellet pure plasma and resuspended in 100 μL of PBS. The concentrations of SEVs estimated by NTA were in the range of 2.1–2.6 × 10^11^ particles/mL, reflecting the individual variability. The sizes of vesicles measured by NTA varied from 85 to 110 nm. A representative example of five averaged measurements (2.4 ± 0.4 × 10^11^ particles/mL) of one sample is presented in Figure 3A.

The morphology of vesicles was analyzed with cryo-TEM. Membrane-formed vesicles of different sizes are visible in Figure 3B; representative images are shown in the enlarged insert. The expression of small extracellular nanovesicles or exosomes markers, i.e., the tetraspanins CD63 and CD9, was assayed by bead-assisted flow cytometry. SEVs were bound nonspecifically to the latex beads, stained with fluorescent-labeled antibodies, and analyzed. Latex beads without SEVs were stained with fluorescent-labeled antibodies in parallel and analyzed as a negative control. As shown in Figure 3C, SEVs isolated by a two-phase polymer system were positive for CD63 and CD9. On the basis of the obtained results, we concluded that plasma SEVs isolated by a two-phase polymer system were present, at least partially, by plasma exosomes. 

#### 3.1.3. Spectral Analysis of the Analytic System Components

The change in absorption spectra in the results of TMB peroxidation induced by AuNPs was supposed as a readout of our analytic system. Before the investigation of the spectral shifts of the analytic system, we tested the spectra of individual components to identify the optical transparency windows. The suspension of AuNPs (260 mg ± 4 mg/mL), suspension of SEVs (6 × 10^10^ particles/mL), and solutions of different ssDNA aptamers specific to CD63, CD30, and PS1NP (25 mM) were prepared, and spectrum analyses were performed. The spectral line shape characteristics for individual components of the analytic system are shown in Figure 4. 

Dominant spectral line peaks of the ssDNA aptamer and SEVs were detected at wavelengths shorter than 300 nm. The AuNP-induced spectral line peak was detected at 520 nm. Thus, any component of the analytic system did not absorb light at wavelength diapasons 330–450 nm as well as >600 nm. The products of TMB peroxidation were expected to have characteristic absorption peaks at 370 and 652 nm; therefore, our system had characteristics suitable for the efficient detection of AuNP-induced TMB peroxidation.

### 3.2. Optimization of the Analytic System Conditions

#### 3.2.1. Interaction of AuNPs and DNA Aptamers

According to the principle of the proposed analytic system, the peroxidase activity of AuNPs should first be suppressed by ssDNA attachment. The quantity of ssDNA aptamers used to suppress the enzymatic activity of AuNPs should be sufficient for complete suppression but not excessive; therefore, free aptamers remain in the reaction mixture. In order to determine optimal reaction conditions, a constant amount of AuNPs (10 µL with concentration 2 × 10^15^ particles/mL) was mixed with 12.5 µL of ssDNA aptamer solution with different concentrations, and incubated for 30 min at 4 °C. The scattered light intensity of the AuNP solution and peroxidase activity of AuNPs were measured in the range of ssDNA aptamer concentrations (Figure 5). 

Thus, unmodified AuNPs had hydrodynamic radii of about 7 nm, as expected (Figure 5A, gold line). Increasing amounts of ssDNA aptamers (5, 10, and 15 mM) induced a slight increase in the AuNP hydrodynamic radii, from 7 to 10 nm (Figure 5A, red lines). This might reflect the process of aptamer adsorption onto the AuNP surfaces. Moreover, the addition of ssDNA to the AuNP suspension resulted in the formation of relatively large particles with hydrodynamic radii of about 100 (5 mM of ssDNA), 140 (10 mM of ssDNA), and 400 nm (15 mM of ssDNA), which might be aggregates of AuNPs induced by free aptamers; however, the exact nature of these particles is still unclear. Taking into consideration the Rayleigh approximation, stating that the intensity distribution is approximately proportional to the size in the power of 6 (R_h_)^6^ of particles [28], the number of these aggregates can be assumed to be small, relative to the number of suspended AuNPs. 

A further increase in the aptamer concentration, up to 20 nM, led to a strong reduction in the mean peak, corresponding to AuNPs in solution, and the appearance of a new peak corresponded to large aggregates with radii >1000 nm (Figure 5A, thick red line). These results indicated a collapse of the analytic system, when ssDNA caused a crucial aggregation of AuNPs. 

In order to assay the suppressive effect of ssDNA aptamer on the peroxidase activity of AuNP, the above-described experiments were repeated in a broader range of ssDNA concentrations. Again, a constant amount of AuNPs was mixed with ssDNA aptamers at different concentrations, incubated (30 min at 4 °C), and then a solution of TMB (10 μL) was added, mixed, incubated (18 min at 37 °C in the dark), and centrifuged (3 min 6000× *g*). The absorption spectrum of supernatant was measured immediately. The results obtained as extinction coefficient characteristics for the product of TMB peroxidation (370 nm and 652 nm) are shown in Figure 5B. As expected, an increase in the ssDNA aptamer concentration resulted in a decrease in AuNP peroxidase activity; a minimum was observed at a concentration of 16.7 mM. Thus, the results of DLS and spectrophotometry were in good agreement, indicating conditions when the concentration of ssDNA (15 mM) was enough to suppress the peroxidase activity of AuNPs; however, it did not cause AuNP aggregation. Interestingly, a further increase in the ssDNA concentration resulted in a slight increase in absorbance, probably associated with the aggregation of AuNPs rather than with their peroxidase activity, as has been reported in [10]. These experiments were performed with ssDNA of different sequences with equivalent results. Therefore, the condition of the AuNP–aptamer formation, i.e., AuNPs (1.6 × 10^15^ particles/mL) and ssDNA (15 mM), was assumed as optimal and was used in further experiments.

#### 3.2.2. Interaction of SEVs with AuNP–Aptamer Complexes

In order to investigate the SEV-sensing property of the analytic system, we used ssDNA aptamers specific to CD63 and maintained constant conditions for the AuNP–aptamer complex formation. The SEVs of the healthy donors were added to the suspension of AuNP–aptamer complexes to obtain a series of final concentrations, i.e., 3.4 × 10^7^, 5.6 × 10^7^, 11.1 × 10^7^, and 16.7 × 10^7^ particles/mL, which were then mixed and incubated for 30 min at 4 °C. First, the composition of particles in the reaction mixture was assayed by DLS. The normalized results are presented in Figure 6.

Suspensions of AuNPs and AuNP–aptamer complexes revealed a dominant peak corresponding to particles with hydrodynamic radii of 7–9 nm. In the mixture of SEVs and AuNP–aptamer complexes, two peaks corresponding to particles with average hydrodynamic radii of 5–9 nm and 26–45 nm were observed. We suggest that the first particle types corresponded to AuNPs, and the second particle type to SEVs. For both types of particles, increasing the SEV concentration to 11.1 × 10^7^ led to a decrease in the average size. A more in-depth analysis is still required to explain this observation. When the concentration of SEVs increased to 16.7 × 10^7^ particles/mL, we registered two fractions of particles with average radii of about 26 nm and >1000 nm, respectively. Thus, an excess of SEVs led to abundant aggregation. In this condition, the light scatter from large particles did not allow us to detect light scattering contributions from small particles such as AuNPs with radii of about 7–11 nm. Importantly, this result indicated the limit of the analytic system’s capacity when an excess of SEVs induced an aggregation of system components and accurate measurements of AuNP peroxidase activity become doubtful. 

Next, we evaluated the shift of peroxidase activity of AuNPs in a series of SEV concentrations with the previously mentioned limit. The complexes of AuNP aptamers were formed at an optimal concentration ratio (AuNPs, 1.6 × 10^15^ particles/mL and CD63-specific aptamers, 15 mM). The SEVs were added up to a final concentration of 16 × 10^7^ vesicles/mL. The suspension of AuNPs (1.6 × 10^15^ particles/mL) without aptamers was used to define the maximum peroxidase activity of the analytic system. The mixtures were incubated (30 min at 4 °C); afterwards, a solution of TMB (10 μL) was added, mixed, incubated (18 min at 37 °C in the dark), and centrifuged (3 min 6000× *g*). Figure 7 shows the results of the experiment. 

After the reaction of TMB peroxidation was completed, we observed a characteristic blue color with gradual intensity reflecting the SEV concentration (Figure 7A). The colorimetry results of the experimental samples revealed growths of peaks at 370 nm (Figure 7B) and 652 nm. Importantly, we observed a strong dependence of optical density from an SEV quantity in a range of concentration from 5 × 10^6^ to 40 × 10^6^ particles/mL (Figure 7C). Thus, the obtained results demonstrated the possibility to measure the concentration of SEVs with the developed AuNP aptasensor. The linear dependency between SEV concentration and AuNP’s peroxidase activity was observed in quite a narrow diapason, i.e., from 5 × 10^6^ to 40 × 10^6^ particles/mL, whereas the observed limit of detection (LOD) was rather average in terms of the parameters reported for aptasensors in other studies [9]. It is now hard to predict whether the reported diapason of the SEV concentration measurement will be sufficient to address any diagnostic questions. This should be determined with many factors, such as the concentration of disease-indicative SEVs, the abundance of measurable vesicular markers, and aptamer performance. Apparently, the practical applicability of AuNP aptasensors should be experimentally explored for any specific clinical task. 

The next experiments aimed to prove whether the developed AuNP aptasensor would be sensitive enough to detect plasma SEVs bearing cHL-specific markers and whether clinically relevant questions could be addressed by the developed system.

### 3.3. Evaluation of the Clinical Performance of AuNP Aptasensor 

#### 3.3.1. Quantification of CD30 + SEVs in the Plasma of cHL Patients and Healthy Donors

We assumed that cHL cells secrete specific SEVs and the development of cHL is associated with the appearance of specific HRS-cell-derived SEVs in circulation. However, many factors in our empirical assumption were unknown, including if selected aptamers would bind specifically cHL cell-derived SEVs among an abundance of plasma vesicles, how representative the fraction of HRS-derived SEVs in the plasma was, and if any background levels of SEVs bind CD30 and PS1NP aptamers in the plasma of healthy donors. The “proof of principle experiment” was a comparison of healthy donors (*n* = 10) and cHL patients (*n* = 10) with newly diagnosed disease. The total populations of SEVs were isolated from pellet pure plasma samples (1.3 mL) by a two-phase polymer system (Section 2.3), the concentration of SEVs in obtained suspensions was in a range of 2.1–2.6 × 10^11^ particles/mL and was not normalized. We used two aptamers against cHL-specific markers (CD30 and PS1NP), one aptamer against common exosomal markers CD63 (as positive control) and scramble aptamer with unknown specificity, SCR (as negative control). The analysis was performed according to the developed protocol (Section 2.11), and absorbance was determined at two wavelengths (370 nm and 652 nm) characteristic for the products of TMB peroxidation. The results are presented in Figure 8. 

The results (mean values and the standard deviation) obtained at 370 nm by the AuNP aptasensor with the CD30 aptamer: cHL patients 0.51 ± 0.07 a.u., donors 0.33 ± 0.03; PS1NP aptamer: cHL patients 0.46 ± 0.05 a.u., donors 0.35 ± 0.04 a.u.; with CD63 aptamer: cHL patients 0.59 ± 0.11 a.u., donors 0.61 ± 0.09 a.u., and with scramble aptamer: cHL patients 0.33 ± 0.02 a.u., donors 0.32 ± 0.03 a.u. (Figure 8A). The results obtained at 652 nm by AuNP aptasensor with CD30: cHL patients 0.31 ± 0.05 a.u., donors 0.21 ± 0.02 a.u.; with PS1NP aptamer: cHL patients 0.28 ± 0.03 a.u., donors 0.21 ± 0.03 a.u., with CD63 aptamer: cHL patients 0.35 ± 0.09 a.u., donors 0.33 ± 0.05 a.u., and with scramble aptamer: cHL patients 0.23 ± 0.04 a.u., donors 0.18 ± 0.01 a.u. (Figure 8B). A statistically significant difference between groups of cHL patients and donors was observed when any of the two HL specific aptamers (CD30 and PS1NP) were. Difference between cHL patients and donors observed in control experiments with CD63 specific and scramble aptamers was not statistically significant and can reflect analytic diapason of method. The results reproduced in measurements at different wavelengths confirmed the analytic system’s reliability. Moreover, the obtained results were the first report of increased CD30 + SEV concentrations in plasma associated with cHL development. Importantly, the observed difference between donors and patients was confidently in the range of AuNP aptasensor’s analytic capacity. Next, we aimed to test whether AuNP aptasensor analytics could be useful for the assessment of clinically relevant characteristics of the disease. 

#### 3.3.2. Comparison of PET-CT Data and CD30 + SEVs Relative Concentration

Positron emission tomography (PET) using 18F-FDG combined with computer tomography is a standard method for the primary assessment of patients with HL and for further monitoring of chemotherapy. The PET-CT analysis was performed for all ten cHL patients included in our study. According to the measured parameters (MTV, metabolic tumor value) and (SUV, standardized uptake value in all separated disease focuses), the complex parameter of the total lesion glycolysis (TLG) was calculated for each patient. This parameter integrally reflects the amount and metabolic activity of disease-affected lymphoid tissue and serves as important criterium for cHL diagnosis. We assumed that the amount of HRS-cell-derived SEVs in plasma is associated with disease severity; therefore, it was expected to correlate with the PET-CT results. 

PET-CT and CD30 + SEV quantification by AuNP aptasensor was conducted for all ten patients enrolled in the study, within an interval of less than 10 days. The correlations between TLG data and the results of AuNP aptasensors composed of CD30, PS1NP and scramble (SCR) aptamers (each measured at two wavelengths) are shown in Figure 9.

The results of PET-CT scanning and the AuNP aptasensor with either CD30 or PS1NP aptamers demonstrated a linear correlation with 95% confidence. No correlation was observed in the control experiments. This result revealed that the amount of HRS-cell-derived SEVs in the plasma of an cHL patient is strongly associated with disease activity, as assayed by PET-CT scanning. Thus, the proposed technology of AuNP aptasensing can be used for the accurate quantification of cHL-derived SEVs and provides clinically relevant information.

#### 3.3.3. Quantification of CD30 + SEVs in the Plasma of HL Patients in the Course of Therapy 

The current algorithm of cHL management involves courses of chemotherapy and, in some cases, followed by radiation therapy [29,30]. The selection of treatment protocols is defined by the stage, subtype of disease, and comorbid medical condition of patients. Considering a relatively high level of cHL curability, the overall goal of cHL management is to define a regime sufficient to treat the disease with minimal toxicity. In this regard, the development of a new supplementary to the PET-CT method for evaluating the chemotherapy response is an important task. 

In order to test the applicability of the AuNP aptasensor for monitoring cHL chemotherapy, we analyzed the plasma from a cHL patient before and after two cycles of ABVD. The representative results, including PET-CT images and CD30 + SEV quantification, are presented in Figure 10.

A female patient, 36 years old, with stage IIA classical HL underwent two cycles of ABVD. As can be seen in the PET-CT images, the volume of pathological focuses of metabolic activity was completely reduced (Figure 10A). In parallel, the analysis of plasma with AuNP aptasensor revealed a considerable decrease in CD30 + SEVs (Figure 10B). The grey line on the graphic almost corresponded to the level of healthy donors which indicated a “normal” level of CD30 + SEVs in circulation. However, the clinical meaning, and perhaps prognostic value of this observation, still require further investigations.

Finally, we quantified CD30 + SEVs in the plasma of the other nine HL patients (total, *n* = 10) before and after therapy. 

Figure 11 shows partial data spectra for 10 patients (before and after chemotherapy) and 10 healthy donors. It can be noted that the quantity of CD30 + SEVs in the plasma of cHL patients decreased considerably after two cycles of chemotherapy. However, the group analysis allowed us to see considerable dispersion within each group. All physiological parameters have a certain reference range; therefore, the results of the quantification of CD30 + SEVs in the group of healthy donors were dispersed in a range of 0.17–0.24 a.u. Comparison of the results of patients with cHL before and after therapy showed a statistically significant difference (black and dark gray spots on the inserted graphs, Figure 11). However, when the results of treated patients and healthy donors were compared (dark gray and light gray spots in the inset plots, Figure 11), the amount of CD30 + SEV in the patient’s plasma appeared to still be elevated. Only one dark grey spot was plotted in a range of the light spot discrepancy intervals; therefore, we can conclude that the measured parameter fell to the “normal” range in only one treated patient. Thus, the results obtained by the AuNP aptasensor revealed different effects of chemotherapy in a group of ten patients that might have a clinically relevant interpretation and/or prognostic significance.

## 4. Discussion

The most important result of our study was the demonstration of the opportunity to quantify disease-indicative small extracellular vesicles in plasma by the AuNP aptasensor based on reversible peroxidase-mimetic activity of AuNPs. The detection of cancer-derived SEVs has become an attractive research topic [31,32]; thus, the presented approach might have great clinical potency; however, some issues with technological and clinical relevancy still need to be addressed. 

### 4.1. The Reasons and Challenges of NP-based Sensor Development

Circulating plasma SEVs present an extremely heterogeneous population. The importance of the structural and functional diversity of extracellular nanovesicles has been stated [33]; therefore, considerable effort has been applied to quantify [34] or even isolate and analyze [35] a tissue-specific population of SEVs. However, the physiological composition of a total population of plasma SEVs has still not been elucidated. Without knowledge of the normal range, it is difficult to quantify or qualitatively assess the disease-induced changes in plasma SEVs. At the current state of understanding, we can just assume that the pathological proliferation and metabolic activity of certain tissues will lead to an increase in tissue-specific vesicles. The portion of tissue- or disease-specific SEVs in a total population of plasma vesicles is rather small; therefore, extremely sensitive sensing technology should be applied. 

Materials at a nanometer scale have unique optical, electronic, enzymatic, and magnetic properties as compared with bulk material, and they have a high surface area to volume ratio, enabling interaction with an enhanced number of biological entities. Therefore, different types of nanomaterial-based sensors are the most promising approach for detecting and analyzing SEVs. Different SEV detection platforms have been reported based on various sensing techniques, including fluorescence, colorimetry, surface plasmon resonance (SPR), surface-enhanced Raman spectroscopy (SERS), electrochemistry, and nuclear magnetic resonance. Different types of target recognition molecules can be used, including antibodies, peptides, lectins, cholesterol anchors, and aptamers. Recently, over forty reports of NP-based sensors for SEVs were comprehensively reviewed and compared [36]. The sensitivity (limit of detection, LOD) of these sensors ranged from 10^2^ to 10^9^ vesicles per mL of solution. However, the vesicles were first isolated from different plasma volumes using different ultracentrifugation protocols or even other methods. This makes it impossible to extrapolate the reported sensitivity rates to clinically relevant units, such as vesicles per mL of plasma or blood. Thus, a true evaluation of various NP-based platforms can be achieved by direct comparison using equal plasma samples and, as far as we know, this has not yet been achieved. Moreover, the intensity of the signals produced by NP-based sensors depends on two factors: the SEV quantity and the density of target molecules on the SEVs’ surfaces. These two factors can have different clinical meanings and interpretations. Therefore, a parallel assessment of SEV samples with an NP-based sensor and independent quantification by another method can help to estimate the vesicular marker expression level and stability of this parameter. The detection of markers constantly and equally expressed on vesicular surfaces would allow one to truly quantify vesicles, whereas detection signals from SEV markers with variable or disease-associated expression should be normalized versus SEV quantity. Thus, further development and clinical implementation of an NP-based sensor for circulating SEVs requires the selection of an optimal platform and evaluation of reliable SEV markers. 

### 4.2. Perspectives of AuNP Aptasensor Application for HL Management

Thanks to multiagent chemotherapy and improved radiation techniques, the majority of patients with limited or advanced stage classical form of Hodgkin lymphoma (cHL) reach complete disease remission by the end of planned treatment [29]. Monitoring the therapy effect in cHL patients with PET-CT scanning has become a standard approach in recent years. Standardization of PET methods and reporting, including the five-point Deauville scale, have enabled the delivery of robust clinical trial data and the development of response-adopted treatment approaches [37]. A PET-CT evaluation of patients after the second cycle of chemotherapy can identify those patients who will benefit from truncated and a less-toxic treatment regime and can omit radiotherapy. However, the group of patients considered for chemotherapy prolongation is quite heterogeneous, and some of these patients will experience either primary refractoriness to the applied drug combinations, or early or late disease relapse. PET-CT scanning is an excellent technique for therapeutic effect assessments, but the activity of the glucose metabolism cannot fully reflect a tumor’s malignant potency, and the predictive value of this parameter is limited. Moreover, PET-CT scanning is associated with radiation exposure and is relatively expensive. Thus, new approaches for disease assessment, as well as monitoring therapy effects and prognoses, are still needed. 

For example, the feasibility and validity analysis of a circulating tumor DNA (ctDNA) in the plasma of cHL patients was evaluated as a new strategy for disease characterization and treatment tailoring [2]. Nine genes commonly mutated in biopsy and ctDNA were identified. It was demonstrated that the detection of these mutations may help to assess early treatment responses associated with KT. Quantification of CD30 + SEVs addresses the same issues, but might have greater clinical potency. ctDNA is a product of cell damage or apoptosis, whereas SEVs are actively secreted by living cells. SEVs secreted by HRS cells may bear a combination of disease-indicative surface markers and might more closely reflect the condition of a tumor. As revealed by the quantification of CD30 + SEVs in the plasma of HL patients before the onset of therapy, the results are closely correlated with the PET-CT data, indicating diagnostic potency of the method. However, the quantification of CD30 + SEVs after two cycles of chemotherapy revealed a certain reduction in the measured parameter. Moreover, the amount of CD30 + SEVs in the plasma of treated patients differed considerably, apparently reflecting different response rates or residual disease activity. It is noted that the therapeutic response evaluated by PET-CT scanning was interpreted as equally completed in all ten patients. The variability of the post-treatment CD30 + SEV evaluation might indicate new clinically relevant disease characteristics that could have prognostic value. 

To apply the proposed technology in clinical practice, we plan to address the following issues: First, the analytic properties of AuNP aptasensors can be improved by using a combination of HRS-specific aptamers. For instance, the development and application of CD15- and/or PAX5-specific aptamers would improve the performance of the diagnostic test. Moreover, it is not necessary to measure absorbance at two different wavelengths and the measurement procedure can be simplified; Second, the approach for further research will be a clinical evaluation of the results of CD30 + SEV quantification. We need to develop an algorithm for AuNP aptasensor result evaluation and clinically relevant interpretation. This will require the estimation of a “normal” range of CD + SEVs in plasma of healthy donors and the elaboration of a method to express dynamic changes in CD + SEV concentrations in individual patients over the course and follow-up of therapy. An analysis of CD30 + SEVs in the plasma of patients with other diseases involving B-cell activation will help to assay the diagnostic specificity of the method. A search for possible correlations of CD30 + SEVs quantity in plasma with other clinically relevant characteristics of cHL patients (B symptoms, blood count, and metabolic activity of residual tumor mass) will estimate the potency of the proposed test as a method for therapy effect monitoring and treatment tailoring. Finally, analysis of HRS-cell-derived SEVs and following up patients with different disease courses will help to reveal the prognostic power of the proposed method. 

## Figures and Tables

**Figure 1 biosensors-12-00023-f001:**
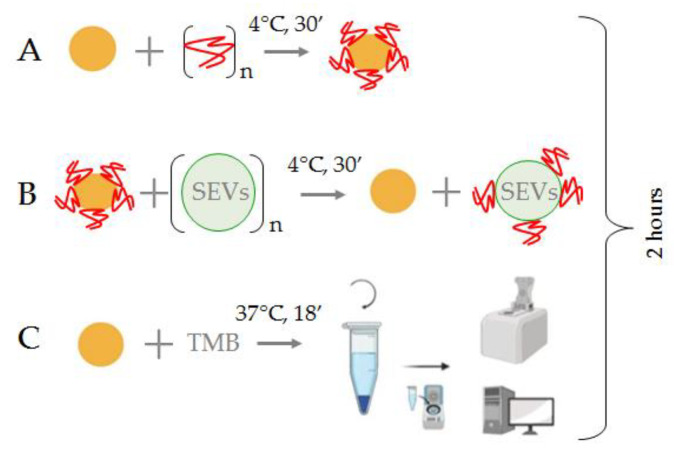
Principal scheme of the SEV quantification with the AuNP aptasensor. (**A**) Formation of the AuNP aptasensor and complete inhibition of NPs enzymatic activity; (**B**) Incubation of the AuNP aptasensor with SEVs, movement of aptamers from the NP surface and recovery of NP enzymatic activity; (**C**) TMB peroxidation and assessment of the intensity of the color reaction.

**Figure 2 biosensors-12-00023-f002:**
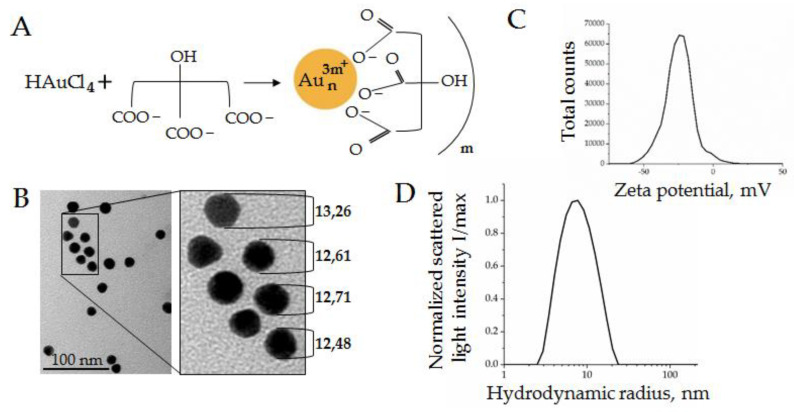
AuNPs characteristics: (**A**) Schema of AuNPs stabilization by citrate solution; (**B**) SEM micrographs of AuNPs; (**C**) ζ-potential measured by a Zetasizer Nano ZS; (**D**) hydrodynamic radius distribution of the normalized scattered light intensity of AuNPs obtained by DLS.

**Figure 3 biosensors-12-00023-f003:**
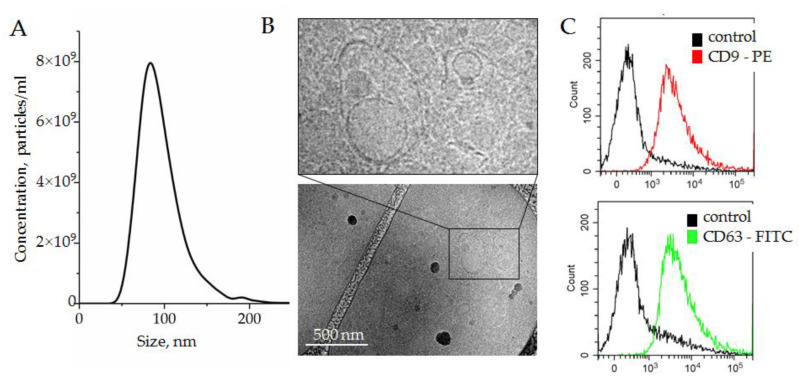
(**A**) Size and concentration of SEVs isolated by the two-phase polymer system from plasma and measured by NTA; (**B**) SEVs visualized by cryo-TEM; (**C**) expression of exosomal surface markers, CD9 and CD63, assayed by bead-assisted flow cytometry. SEVs were attached on the surface of latex beads by nonspecific electrostatic interactions, and then stained with fluorescent-labeled antibodies against either CD9 or CD63. Latex beads without SEVs were stained with antibodies and assayed as a negative control.

**Figure 4 biosensors-12-00023-f004:**
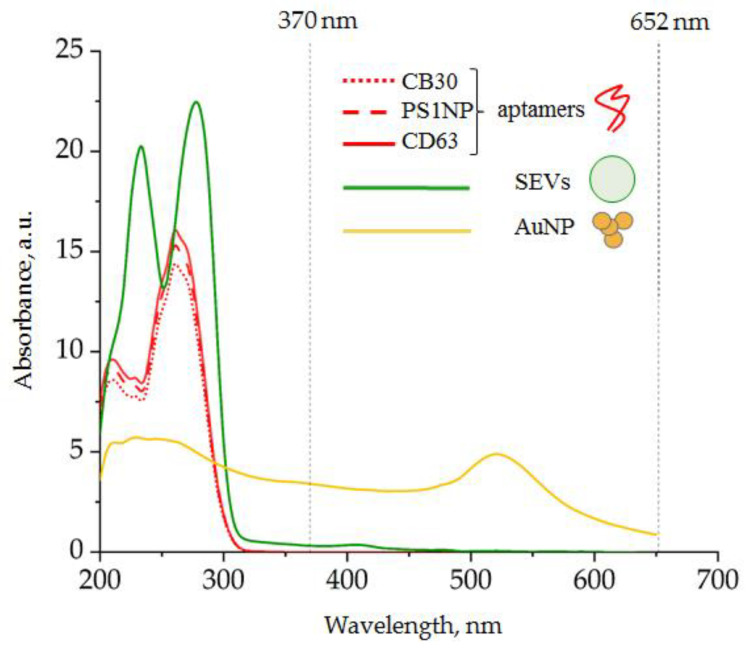
Spectra of individual components of the analytic system: (red) DNA aptamer, 25 mM; (green) SEVs, 6 × 10^10^ particles/mL; (yellow) AuNPs, 260 mg ± 4 mg/mL.

**Figure 5 biosensors-12-00023-f005:**
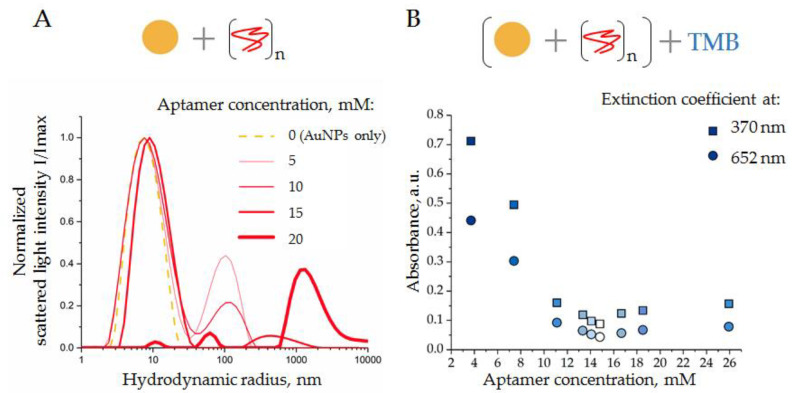
(**A**) DLS distributions of AuNP in the presence of ssDNA aptamers of different concentrations (see the legend on the graph); (**B**) activity of TMB peroxidation by AuNPs in the presence of ssDNA aptamer, absorbance at two wavelengths (370 and 652 nm). Schemes of the experiments are displayed above the graphs.

**Figure 6 biosensors-12-00023-f006:**
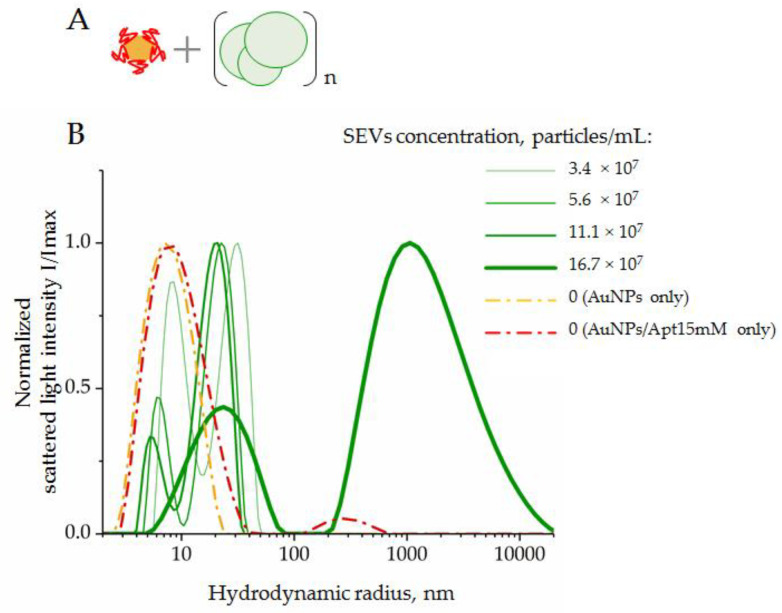
DLS distribution of hydrodynamic radius, R_h_, of the system components composed by AuNPs complexed with CD63 ssDNA aptamers in a range of SEV concentrations: (**A**) scheme of the experiment; (**B**) results of DLS measurements. AuNP–aptamer complexes were formed at AuNPs (2 × 10^15^ particles/mL), and an aptamer concentration of 15 mM was assayed as control.

**Figure 7 biosensors-12-00023-f007:**
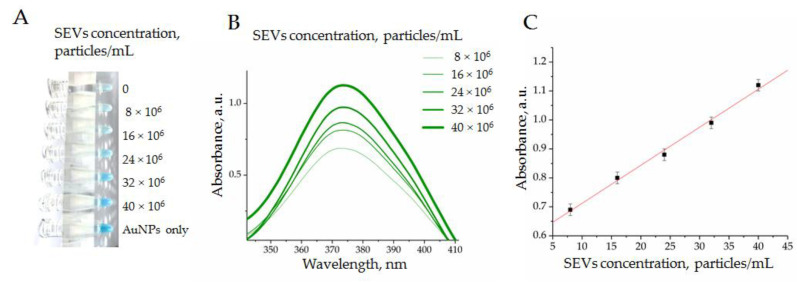
Enzymatic activity of AuNPs complexed by CD63 ssDNA aptamers in a range of SEV concentrations evaluated by TMB peroxidation: (**A**) the photography of tubes; (**B**) absorbance measured in experimental probes at 370 nm; (**C**) dependency of absorbance from SEV concentration in a range from 5 × 10^6^ to 40 × 10^6^ particles/mL.

**Figure 8 biosensors-12-00023-f008:**
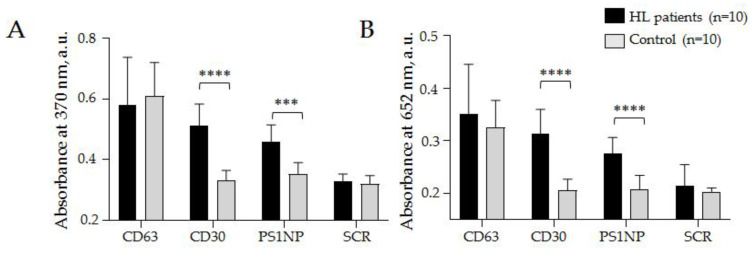
Relative quantification of CD30 + SEVs in the plasma of HL patients and healthy donors. CD63-specific (CD63) and scramble (SCR) aptamers were used in parallel as positive and negative controls, respectively. The absorbance was estimated at wavelengths of 370 nm (**A**) and 652 nm (**B**) and averaged within clinical groups. The statistical significance of the observed difference was evaluated with the nonparametric Mann–Whitney test (*** *p* < 0.0005 and **** *p* < 0.00005).

**Figure 9 biosensors-12-00023-f009:**
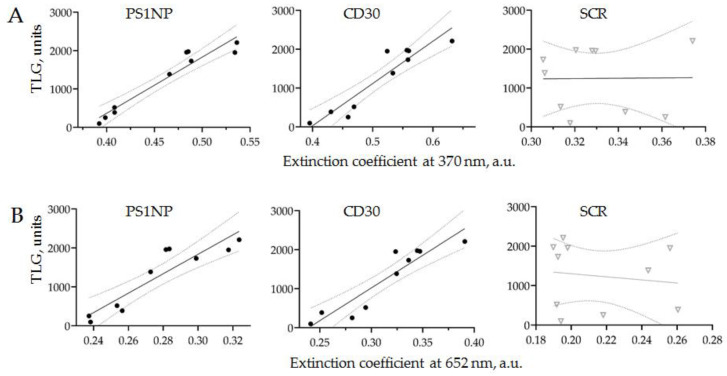
The correlation of the PET-CT results (TLG) and the assessment of HRS-cell-derived SEVs in the plasma of cHL patients using AuNP aptasensor. HRS-cell-derived SEVs were quantified using three types of aptamers: CD30, PS1NP and SCR as controls, and absorbance was measured at two wavelengths (370 and 652 nm).

**Figure 10 biosensors-12-00023-f010:**
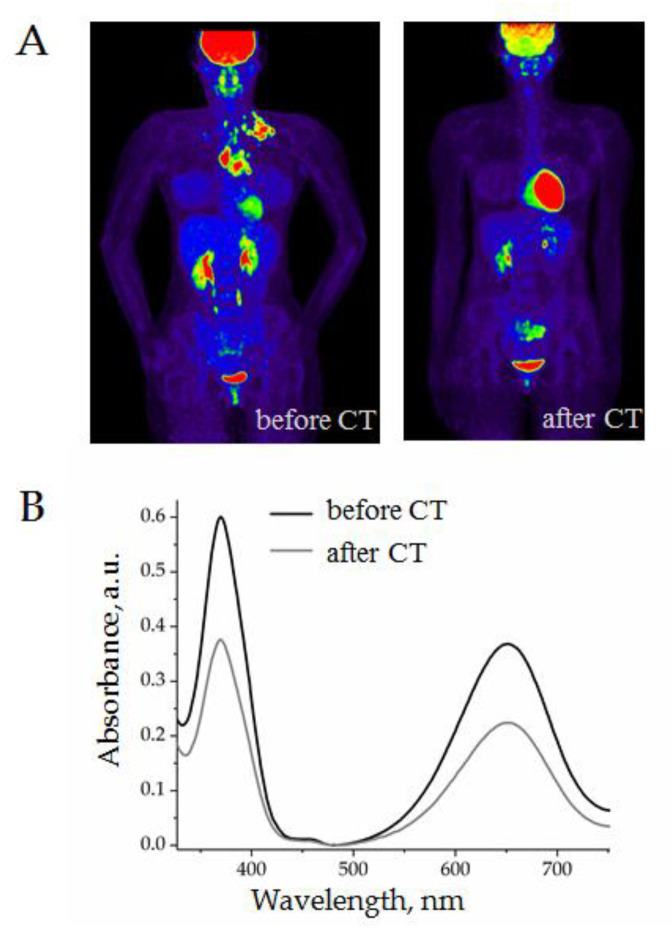
Evaluation of the results of two cycles of chemotherapy (ABVD) of a 36-year-old female patient: (**A**) PET-CT images before and after treatment; (**B**) quantification of CD30 + SEVs in plasma by the AuNP aptasensor before and after treatment.

**Figure 11 biosensors-12-00023-f011:**
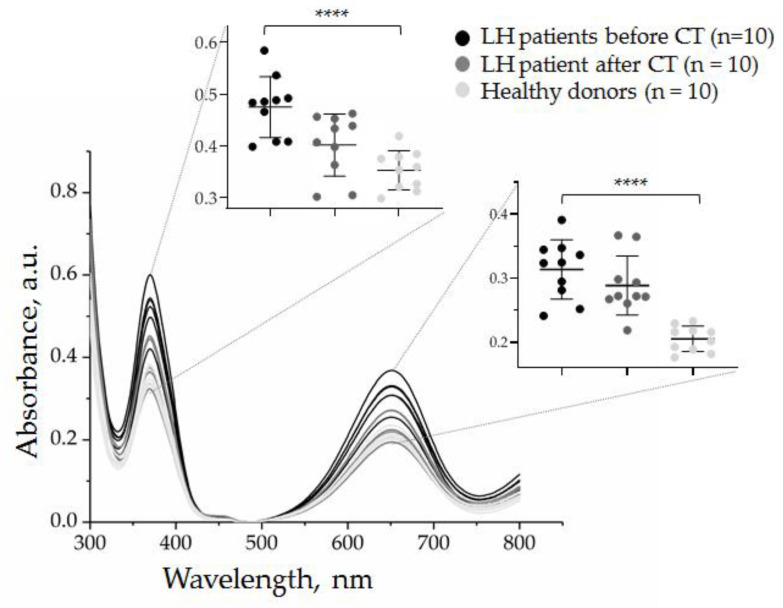
Evaluation of the effect of two cycles of chemotherapy on the quantity of CD30 + SEVs in the plasma of cHL patients by the AuNP aptasensor with the CD30 aptamer. The results of the analysis of ten samples of healthy donors are included for comparison. The results of the absorbance measurements at 370 and 652 nm are presented in inserts. The statistical significance of the observed difference was evaluated with the nonparametric Kruskal–Wallis test (**** *p* < 0.00005).

**Table 1 biosensors-12-00023-t001:** Patients’ characteristics.

Patient ID	Gender/Agee	Morphology *	Stage	B Symptoms	ECOG	Before Chemotherapy		After Chemotherapy		Deauville
						MTV, cm^3^	TLG	MTV, cm^3^	TLG	
577848	M/33	MCHL	4	+	1	563	1975	277	507	2
583352	F/29	MCHL	2	-	0	72	254	0	0	2
525340	M/30	NSHL	4	+	1	1033	2212	0	0	2
586446	F/29	NSHL	3	+	2	310	1731	0	0	3
585071	M/44	NSHL	4	-	1	251	1387	26	50	2
590335	F/42	NSHL	2	-	0	91	520	0	0	2
589465	M/54	MCHL	4	+	1	400	1959	0	0	3
596063	M/27	NSHL	2	-	1	356	1951	0	0	2
438920	M/29	NSHL	2	-	0	47	99	0	0	2
598035	F/36	NSHL	4	-	1	157	388	0	0	3

* NSHL, nodular sclerosis Hodgkin lymphoma; MCHL, mixed cellularity Hodgkin lymphoma.

## Data Availability

Data are available upon request.

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
