# Peer review of "AuNP Aptasensor for Hodgkin Lymphoma Monitoring"

_biosensors, 2022, doi:10.3390/bios12010023_

Round 1

Reviewer 1 Report

  1. The abstract part lacks the principle of aptamer sensor, sensing process, color rendering process, detection range.
  2. In Table 1, there are 3 cases of mixed cellularity HL, which does not meet the "revealing two cases of mixed cellularity HL and eight cases of nodular sclerosis 249" in the main text.HL".
  1. In section “3.1.3. Spectral Analysis of the Analytic System Components”, ssDNA aptamer specific to CD63 (25 mM) was used to spectrum analysis. But the target is CD30 + SEVs. Why not ssDNA aptamer specific to CD30.
  2. What is the meaning of ENVs in Figure 4, Figure 6 and Figure 7

Author Response

Dear Reviewer! Thank you very much for careful analysis of our manuscript and helpful suggestions!

  1. The abstract part lacks the principle of aptamer sensor, sensing process, color rendering process, detection range.

The principle of sensor, sensing process and readout are now reflected in the abstract. As detection range is mediated by complex of factors (the aptamer properties, the detected marker expression / amount of aptamer docking sites on single SEV surface, the specific SEVs quantity) and is rather discussion point, we decided to not include this information into the abstract. This issue was explicitly discussed in Discussion section.

  1. In Table 1, there are 3 cases of mixed cellularity HL, which does not meet the "revealing two cases of mixed cellularity HL and eight cases of nodular sclerosis 249" in the main text HL".

Thank you for this remark. The mistake was corrected.

  1. In section “3.1.3. Spectral Analysis of the Analytic System Components”, ssDNA aptamer specific to CD63 (25 mM) was used to spectrum analysis. But the target is CD30 + SEVs. Why not ssDNA aptamer specific to CD30.

As nucleic acids absorb light at wavelengths of 260 / 280 nm due to the resonance structure of the pyrimidine and purine bases, the spectral characteristics of ssDNA aptamers used in the study were expected to be equal. That way we did not include all of them spectra in the Figure 4 and in section 3.1.3.; the section was limited by the description of the aptamer (CD63) used for further system validation. In according to your suggestions, lines reflected spectra of other aptamers are included in the Figure 4.

  1. What is the meaning of ENVs in Figure 4, Figure 6 and Figure 7

Thank you for this comment. The inconsistency (ENV vs SEV) resulted from multiple revision of the manuscript by many co-authors.  Errors are corrected.

Reviewer 2 Report

Slyusarenko and coworkers present a novel method of monitoring disease activity in (classical) Hodgkin lymphoma (cHL) by detecting CD30+ small extracellular vesicles (SEVs). cHL patients had higher levels than controls, there was a strong correlation with TLG, and decreasing but not normalized levels were found after 2 courses of chemotherapy.

The number of study subjects (10 patients and 10 controls) limit the power of any statistical analyses, but is reasonable in a pilot study of such a novel technology.

Comments:

  • The authors write "... the classical form of Hodgkin lymphoma ..." in the first line of the introduction, which is good. I suggest using the term "classical Hodgkin lymphoma (cHL)" throughout the paper.
  • A few typos, like H-SR instead of H-RS cells should be corrected. Please proof read once more. Also, HRS should be preferred over H-RS.
  • The description of PET/CT procedure is ambitious and seems well done, including TMTV and TLG in addition to Deauville scoring. Further, the biopsies seems to have been re-evaluated according to histology (MC, NS). However, as I understand, there is no pathologist or radiologist in the author list. That is of course ok, but some hint of whether all this was done in clinical routine or specifically for the study and in such a case, by whom, should be added.
  • Some wordings could be improved, e.g. 

    "the volume of pathological focus of metabolic activity was reduced completely". 

  • It would be interesting to know if the CD30+ SEVs decreased in all patients after 2 courses of chemotherapy. I believe we now see all individuals before and after and the controls, but not the change on an individual level. This would give us a hint of the usefulness as a treatment evaluation. I suggest to at least add how many of the patients had a decreased level or, even better, couple the results of each individual.
  • It would also be of interest to see these results correlated to TLG, for the 9 patients in figure 11 (since the TLG data is known). If some patient didn't have a decreased level, did that correlate to poorer PET-response?

Author Response

Dear Reviewer!  Thank you very much for careful analysis of our manuscript and helpful suggestions!

1. The authors write "... the classical form of Hodgkin lymphoma ..." in the first line of the introduction, which is good. I suggest using the term "classical Hodgkin lymphoma (cHL)" throughout the paper.

Suggestion is accepted, “HL” replaced by “cHL” throughout the text.

2. A few typos, like H-SR instead of H-RS cells should be corrected. Please proof read once more. Also, HRS should be preferred over H-RS.

Thank you! Errors are corrected.

3. The description of PET/CT procedure is ambitious and seems well done, including TMTV and TLG in addition to Deauville scoring. Further, the biopsies seem to have been re-evaluated according to histology (MC, NS). However, as I understand, there is no pathologist or radiologist in the author list. That is of course ok, but some hint of whether all this was done in clinical routine or specifically for the study and in such a case, by whom, should be added.

Thank you very much for this comment. We worked together with our colleagues from department of radionuclide diagnostics (Dr. Alina Valitova and Dr. Pavel Krzhivitsky), they are both included in author list and now their input is reflected in Author Contribution paragraph, as well as hematologists worked in our team (Dr. Sergey Shalaev and Dr. Larisa Filatova; Dept. of Hematology and Chemotherapy). We used result of routine morphological evaluation of lymph node biopsies, therefore we did not include pathologists in author list. Now input of pathologist is emphasized in Acknowledgment paragraph.

4. Some wordings could be improved, e.g. "the volume of pathological focus of metabolic activity was reduced completely". 

Mistake is corrected. Thank you!

5. It would be interesting to know if the CD30+ SEVs decreased in all patients after 2 courses of chemotherapy. I believe we now see all individuals before and after and the controls, but not the change on an individual level. This would give us a hint of the usefulness as a treatment evaluation. I suggest to at least add how many of the patients had a decreased level or, even better, couple the results of each individual.

Thank you for this important comment.

We agree, the individual data are necessary to evaluate effect of therapy. However, right now we are not able to propose any confident way to interpret these results, that way we decided do not include them into manuscript. For example, we could calculate residual signal as a percentage from initial signal (the way similar to response evaluation criteria in solid tumors, RECIST) for each patient (for 370 nm only, as example). For group of 10 patients included, this numbers (%) are: 90,4; 113,6; 80,8; 93,4; 99,3; 97,5; 62,4; 69,7; 62,0; 89,0. However, this approach does not seem to be correct, because baseline level is not zero. If we would average results of healthy donor and consider this number as a baseline level, results became even less reasonable: 64,8; 218,4; 43,9; 76,2; 97,1; 81,4; -39,2; 23,4; -34,2; 19,1%. Obviously, averaging of results from 10 individual cannot truly reflect “reference” as well as above shown methods of results evaluation are not satisfactory.

In order to address your suggestion, we added additional paragraph in Discussion section and submitted raw data as a supplementary Table 1 (Figure 11_raw data). However, we prefer to keep a methodology as a main focus of presented study. Now we are collecting the data from large cohort of patients and working together with our haematologists to identify any correlation between AuNP-aptasensor data and clinical parameters as well as to find optimal approach of AuNP-aptasensor data evaluation. For instance, a most intriguing issue was a very different response to 2 cycles of chemotherapy revealed by AuNP-aptasensor when result of PET-CT demonstrated almost “complete response” in almost all patients. Our next report will definitely include more clinically relevant results.

6. It would also be of interest to see these results correlated to TLG, for the 9 patients in figure 11 (since the TLG data is known). If some patient didn't have a decreased level, did that correlate to poorer PET-response?

The response to this question is very relevant to previous one. As it was mentioned above, the discrepancy between the therapy effect evaluation by PET-CT and AuNP-aptasensor was observed. We can suppose that AuNP-aptasensor indeed reflected amount of circulating HRS-derived SEVs and can provide us with important and clinically relevant information in addition to PET-CT data. However, we need to expand this research and to attract more haematologists to our team to address your question fully and to conduce this study successfully.

Reviewer 3 Report

The paper “AuNP Aptasensor for Hodgkin Lymphoma Monitoring” by M. Slyusarenko and coauthors describes the development of the novel analytical approach to detect Hodgkin lymphoma (HL)-associated extracellular vesicles, based on DNA aptamer as sensing and bound element and Au nanoparticles as signal element of the system due to  peroxidase-mimicin activity of these particles. The work is extremely relevant, since modern diagnostic technologies for diagnosing this disease are complex and often unavailable. The proposed approach is original. The authors conducted extensive and comprehensive experimental studies to identify the optimal analytical conditions. The results obtained are statistically confirmed and do not raise doubts. Patient studies are of particular interest as far as the proposed analytical approach has shown good prospects for clinical use.

One remark:

In mns text the authors use the abbreviation SEV (small extracellular vesicles), while in Figures 1 and 4 – ENV (extracellular nanovesicles). As far as I can understand they are synonyms here, aren’t they? It would better to specify it.

The text reviewed contains several fragments highlighted in color, it is not clear why.

And a question:

Is it possible to measure not the entire absorption spectrum, but the optical density at only one wavelength (370 or 652 nm) to simplify the analysis?

To summarize, the work is undoubtedly of interest for specialists and can be accepted for the publication after a small revision.

Author Response

Dear Reviewer! Thank you very much for careful analysis of our manuscript and the remarks.

  1. In mns text the authors use the abbreviation SEV (small extracellular vesicles), while in Figures 1 and 4 – ENV (extracellular nanovesicles). As far as I can understand they are synonyms here, aren’t they? It would better to specify it.

Yes, of course, they are synonyms. The inconsistency (ENV vs SEV) resulted from multiple consequent revision of text by several co-authors. We correct it now and kept equal abbreviation (SEV) throughout all text and figures.

  1. The text reviewed contains several fragments highlighted in color, it is not clear why.

As we had some suggestions from scientific editor of Biosensors, revised and added fragments were highlighted. Now these color indications are removed.

  1. Is it possible to measure not the entire absorption spectrum, but the optical density at only one wavelength (370 or 652 nm) to simplify the analysis?

Absolutely. We include this results in the manuscript with only purpose to demonstrate better the method developed. In further work we kept results obtained at 370 nm only because they seem to have a greater spread between individual samples. In practice, one type of measurement will be well enough.

Round 2

Reviewer 1 Report

The authors have revised all the comments. It can be accepted now.